# Multi-Band Analogue Electromagnetically Induced Transparency in DoubleTuned Metamaterials

**DOI:** 10.3390/nano11112793

**Published:** 2021-10-21

**Authors:** Wei Huang, Ningye He, Renxia Ning, Zhenhai Chen

**Affiliations:** 1State Key Laboratory of ASIC and System, Shanghai Institute of Intelligent Electronics & Systems, School of Microelectronics, Fudan University, Shanghai 200433, China; eehuangw@fudan.edu.cn; 2School of Information Engineering, Huangshan University, Huangshan 245041, China; hny@hsu.edu.cn (N.H.); chzh@hsu.edu.cn (Z.C.); 3Engineering Technology Research Center of Intelligent Microsystems, Huangshan 245041, China

**Keywords:** analogue electromagnetically induced transparency, multi-band, tuned, graphene, liquid crystal

## Abstract

A multi-band analogue electromagnetically induced transparency (A-EIT) metamaterial is proposed. The structure is composed of liquid crystal (LC) layer and a graphene strips layer on both sides of silicon dioxide. The transmission spectrum and electric field distribution of only one graphene strip and two graphene strips have been studied. As a bright mode, the graphene strip is coupled with adjacent graphene strip to realize the A-EIT effect. When multiple graphene strips are coupled with each other, the multi-band A-EIT is obtained due to the electric dipole resonances of the four strips. The results show that the multiband A-EIT effect can be tuned by voltage on LC and graphene layer, respectively. Moreover, changing the incident angle of the electromagnetic wave has had little influence on the transmission window in the low frequency band, it is meaning that the A-EIT effect with insensitive to the incident angle can be obtained. Each transmission window has a high maximum transmittance and figure of merit (FOM). The multi-band A-EIT effect can widen the application on sensor and optical storage devices.

## 1. Introduction

Metamaterials are a kind of materials whose subwavelength structures produce properties that natural materials do not have through periodic or aperiodic arrangement. The properties of these materials depend on the artificial structure rather than the material itself [1,2]. It has potential applications in electromagnetic stealth devices, electromagnetic absorbers and electromagnetic sensors. Once the structure of traditional metamaterials is determined, its characteristics will be fixed, which limits its flexibility in application. Therefore, researchers began to study tunable metamaterials. As typical electrically tunable materials, liquid crystal (LC) and graphene can be flexibly adjusted in the metamaterials [3,4].

As a kind of anisotropic material, LC has birefringence effect in terahertz (THz) band. LC molecules will reorient under the action of external electric or magnetic field, and the permittivity shows good continuous adjustable ability in a wide range. LC was widely studied as functional devices in the THz band [5]. R. Kowerdziej et al. proposed anematic liquid crystal (NLC)structure to determine the influence of alternating current (AC) bias voltage on the metamaterial sensor [6]. A sensor was composed of aLC tunable filter and automaticbackground signal subtraction. The results show that the sensor can realize remote detection by software control [7].

A special physical phenomenon of non-linear effect was produced in the process of interaction between light and matter. After decades of research and development, electromagnetically induced transparency (EIT) has faded its mysterious veil [8]. Nevertheless, some of its characteristics, such as slowing light and optical storage and Kerr effect, remain fascinating [9]. The phenomena ofanalogue-EIT (A-EIT) was found in the circuit system [10] and the waveguide coupled system [11]. The study of A-EIT has been widely spread. Among them, the A-EIT effect of the metamaterials has been widely concerned in the microwave [12], visible [13], terahertz [14] and infrared bands [15].

Graphene metamaterials (GMs) have the characteristics of fast response, electrical tunability and ultra thin. GMs have potential applications in electromagnetic lens, electromagnetic absorption and so on. Cheng et al. realized the active and adjustable EIT phenomenon in mid infrared band by using periodic graphene nanoband [16]. A planar metamaterial of A-EIT was experimentally investigated. The electrical size of the metamaterial structure is only 0.2*λ*_0_. The measurement result was shown a high quality factor transmitted peak [17]. Xiang etal.designed an EIT structure of H-shaped GMs. When the symmetry of H-shaped structure is broken, a transparent window can be observed. With the increasing of asymmetry, the width of the transparent window will increase, and the EIT phenomenon is more obvious [18].

A graphene metasurface structureis composed of mono-layer graphenenano-disks coupled with graphene nano-strip exhibited tunable EIT responses at a mid-infrared range [19]. The all-dielectricmetasurface structure was investigatedandcanbe a dynamic manipulated EITeffectin theoptical telecommunication frequencies. The modulation of the EIT window can engender a modulation depth of 88% [20]. The EIT structure composed of graphene strip and a pair of SRR was designed. A tunable EIT can be observed in the THz band, and the change of equivalent parameters of EIT system with the change of relative position is explained by using two particle model [21]. The single tuned EIT effect has been widely reported, however, the double tuned EIT effect has received less attention [22].

In this paper, a three-layer metamaterial is studied, which is composed of a liquid crystal layer, a substrate layer and a graphene layer. The A-EIT phenomenon of double electric tuning is found by the finite integral time domain method. By analyzing the distribution of electric field and magnetic field, the physical mechanism of A-EIT effect is obtained. Through the coupling of several graphene strips, the multi band A-EIT effect is obtained. The results have potential applications in THz devices.

## 2. Geometric Model

The LC is an anisotropic medium, the refractive index of the LC *n_eff_* is given by Equation (1) to the incident THz wave [23].
(1)$$n_{eff}=\frac{n_{e}n_{o}}{\sqrt{n_{0}^{2}\cos^{2}\theta +n_{e}^{2}\sin^{2}\theta }},$$
where *n*_0_ and *n*_e_ represent the ordinary and the extra-ordinary indices, respectively. It is known that the LC is a nonmagnetic material. Thus, *ε_eff_* = *n_eff_*^2^. Figure 1 shows that the *ε_eff_* and *n_eff_* varied with bias voltage (rotation angle of liquid crystal, *ϕ* = 0° denoted the unbiased voltage and *ϕ* = 90° denoted the fully biased voltage).

Assuming that the electronic band structure of a graphene sheet is not affected by the neighboring, so the effective permittivity *ε**_G_* of the graphene be written as [24] follows:(2)$$\varepsilon_{G}=1+\frac{\sigma_{G}}{t_{G}\omega \varepsilon_{0}}$$
where, *σ_G_* is the surface conductivity of the graphene, *t_G_* is the thickness of graphene sheet, *ε*_0_ is the permittivity in the vacuum.

For a graphene sheet, the electromagnetic properties are described in terms of the surface conductivity *σ_G_* whichcan be taken into account in inter-band and intra-band transitionsby the Kubo model of conductivity [25].
(3)$$\sigma_{G}=\sigma_{\mathrm{inter}}+\sigma_{\mathrm{intra}},$$
(4)$$\sigma_{\mathrm{inter}}=\frac{ie^{2}k_{B}T}{\pi \hbar (\omega +i/\tau )}(\frac{\mu_{c}}{k_{B}T}+2\ln(e^{-\frac{\mu_{c}}{k_{B}T}}+1)),$$
(5)$$\sigma_{\mathrm{intra}}=\frac{ie^{2}}{4\pi \hbar }\ln\left|\frac{2\mu_{c}-\hbar (\omega +i/\tau )}{2\mu_{c}+\hbar (\omega +i/\tau )}\right|,$$
where, *ω* is radian frequency, *ħ* is the reduced Planck constant, *κ**_B_* is the Boltzman constant, *e* is the charge of an electron, *T* is the temperature, *μ*_c_ denotes the chemical potential, and *τ* is electron-phonon relaxation time, respectively.

The effective permittivity of graphene *ε**_G_* can be tuned by the *µ_c_* is shown in Figure 2. It is shown that the real part of *ε**_G_* increases with the decrease of *µ_c_*. The change of imaginary part of *ε**_G_* is just the opposite.

The proposed metamaterial structure was calculated by applying the finite integral time domain (FITD) method. The boundary condition ofx and y directionsare unit cell and the *z*-axis is open. The proposed structureconsists of graphene strip and LC layer patches on the opposite sides of polyimide layer of substrate in Figure 3. The structure parameters are as follows: *l* = 18 µm, *h* = 15 µm, *t*_1_ = 1.6 µm, *t*_2_ = 0.3 µm, *d_G_* = 1 nm, *w* = 0.3 µm, *l*_1_ = 4.4 µm, *l*_2_ = *l*_1_ × 1.4 µm, *l*_3_ =*l*_1_ × 1.4^2^ µm, *l*_4_ = *l*_1_ × 1.4^3^ µm.

## 3. Results and Discussions

Figure 4 shows that the transmission spectrum varied with different *ε*_LC_. It is demonstrated that the transmission spectrum changing due to effective permittivity of LC can be tuned by the different voltage on the LC. The transmission spectrum has a slight red shift atthe *ε*_LC_ increases from 2.47 to 2.91. The transmission spectrum produces a larger shift when the *ε*_LC_ increases to 3.06.

Changing the chemical potential of graphene *µ_c_* can adjust the transmission spectrum. As shown in Figure 5, the A-EIT window changes with the *µ_c_* from 0.1 eV to 0.8 eV. Increasing the *µ_c_* causes the A-EIT window to besignificant blue shifted. The resultsshow that the chemical potential of graphene can modulate the transmission spectrum.It can be concluded that the structure has the double tunableproperty.

To study the A-EIT phenomenon of the designed structure, we study separately the Graphene Strips 1 and 2 and both strips inthe A-EIT structure to clarify the cause of A-EIT effect physical mechanism. Asshown in Figure 6b, it is obvious that the transmissionof the Graphene Strips 1 and 2 as bright modes can be excited by the incident wave. Therefore, the bright–bright mode couple to each other cause the A-EIT effect. Theperiodic unitof the metamaterial structure consists of two graphene strips with different lengths. The two graphene strips act similar to radiation antennas, which can be coupled directly with the incident wave and produce the bright mode resonance. Because the length of the two graphene strips is close and the resonance frequency is similar, based on the weak coupling effect of the two bright modes, the metamaterial structure produces a strong A-EIT, and brings a high group refractive index near the transparent window.

To further explain the physical mechanism of the A-EIT, Figure 6a,c show that the A-EIT effect is produced by the coupling of magnetic resonance and electric resonance. It can be seen from Figure 6d that the electric field power in the low frequency band is mainly distributed on Graphene Strip 2. Figure 6e shows the electric field power distribution at the transparent frequency, where the electric field distribution is very weak, indicating that transparency occurs at the frequency of 8.25 THz. Figure 6f shows the electric field distribution of the resonance point in the high frequency band. The energy is mainly concentrated on Graphene Strip 1, indicating that the resonance frequency is determined by Graphene Strip 1 [26].

Figure 7a shows the transmission spectrum of Graphene Strips 1, 2, 3, and 4, respectively. The transmission spectrum has a significant red shift due to the increase of length of the graphene strip. In Figure 7b, four graphene strips are placed in parallel, and weak coupling will occur between them, which results in many strong dispersion transparent curves of A-EITdue to the electric dipole resonances of the four strips. It can be concluded that the multi band A-EIT effect is due to the coupling between multiple bright modes.

Figure 8 shows the electric field distribution of each frequency point in Figure 7b. It can be seen from Figure 8a that the resonant frequency valley a is mainly generated by the coupling between the bright mode Graphene Strips 4 and 3. Figure 8b shows the electric field distribution of the Transmission Peak b, where the energy is mainly concentrated on Graphene Strip 3. Figure 8c–g show the similar results that the energy coupled with neighbouring graphene strip. 

Figure 9 shows the effect of changing *l*_1_ on the transmission spectrum. It can be seen that the transmission spectrum has a red shift with the increase of *l*_1_. This is due to the length of graphene strip increasing, its resonant frequency will also produce a red shift. This conclusion is consistent with the result in Figure 7a.

Figure 10 shows the variation of A-EIT window on oblique incidence. It is shown that the resonance of low frequency band hardly changes with the incidence angle. The band of A-EIT window narrows with the increase of the incidence angle. The high frequency band was produced as a red shift at the certain range. A new resonance frequency can be obtained on the large incidence angle. The above results may be due to the fact that the equivalent dielectric constant of the structure does not change with the incident angle in the low frequency band, while the high frequency band produces a red shift with the increase of the incident angle, resulting in different results of the change of the incident angle in different frequency bands [27].

Figure 11 shows that the transmission spectrum varied with the refractive index of the surrounding environment. With the increase of the refractive index of the surrounding environment *n*_b_, each resonant frequency has a slight red shift. This result has a potential application in refractive index sensing.

To further discuss the application on refractive index sensing, Figure 12a shows the frequency shift at different resonance frequency. It can be seen that the frequency shift of frequency *f*_a_, *f*_c_, *f*_e_, is smaller and the frequency shift of *f*_g_ is larger. Figure 12b shows the quality factor (Q) and FOM varied with the refractive index *n* of the surrounding environment, which is between 13 and 18.

To compare the results of this work with reported, Table 1 shows the multi band A-EIT effect in different bands. It is found that most tunable A-EIT effects can only achieve single tuning. In this study, a dual tunable A-EIT is realized by using LC and graphene. The multi band A-EIT window generated and FOM can reach 18. These results have potential applications on refractive index sensors.

## 4. Conclusions

In this paper, we studied a metamaterial structure with multi-band A-EIT effect. The structure is consisting of LC-SiO_2_-graphene strips layer. To study the A-EIT effect, we first designedthe structure with only a transmissionwindow of A-EIT, which is composed of two graphene strips and LC on both sides of SiO_2_ layer. The transmission spectrum and electric field distribution of only one graphene strip and two graphene strips have been studied. As a bright mode, the graphene strip is coupled with adjacent graphene strip to realize the analogue A-EIT effect. The multi band analogue A-EIT is obtained on multiple graphene strips which are coupled with each other. The results showthat the multibandA-EITeffectcan be tuned by voltage on the LC and graphene layers, respectively. The results show that the incident angleinsensitive A-EIT can be obtained. Moreover, each transmission window can achieve high transmittance and FOM on changing the surrounding environment. The multiband A-EIT effectcan be applied on optical storage devices, sensor and large angle devices.

## Figures and Tables

**Figure 1 nanomaterials-11-02793-f001:**
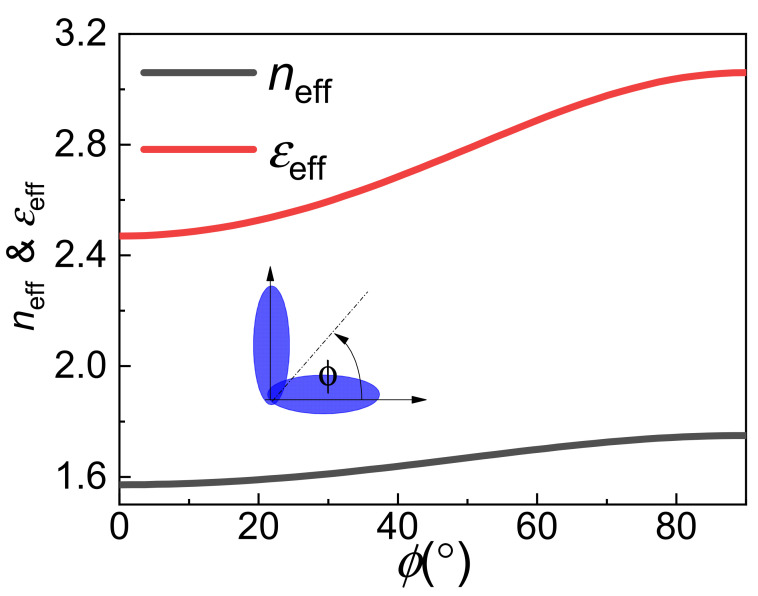
Effective refractive index (*n_eff_*) and permittivity (*ε**_eff_*) varied with rotation angle *ϕ* of liquid crystal.

**Figure 2 nanomaterials-11-02793-f002:**
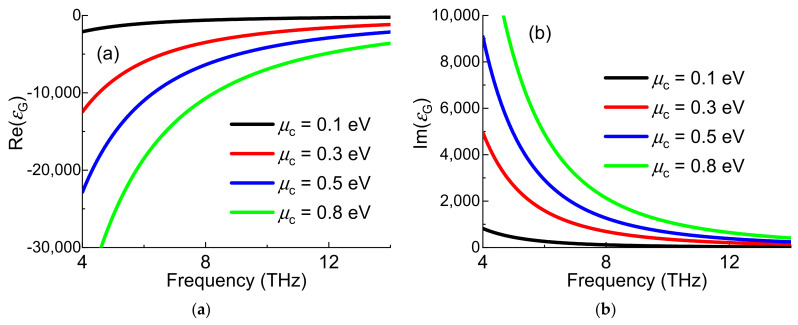
The (**a**) real part and (**b**) imaginary part of effective permittivity of graphene varied with frequency.

**Figure 3 nanomaterials-11-02793-f003:**
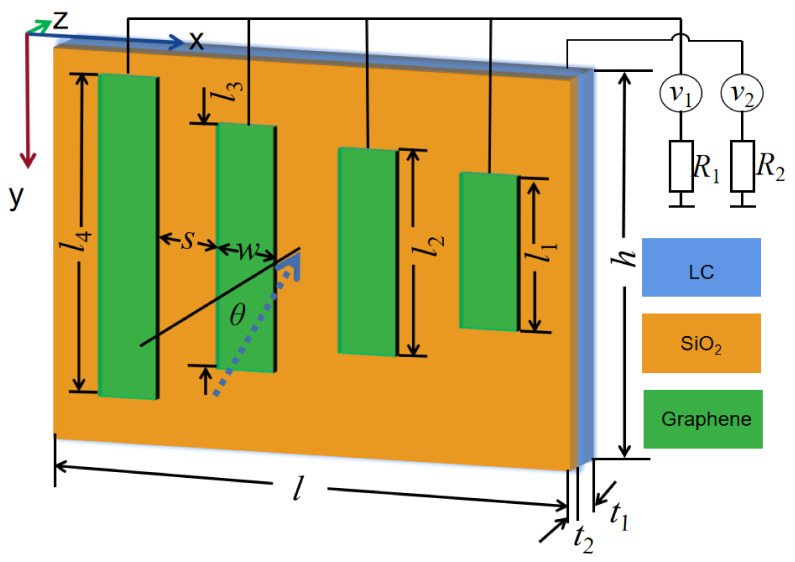
The schematic illustration of our proposed EIT structure.

**Figure 4 nanomaterials-11-02793-f004:**
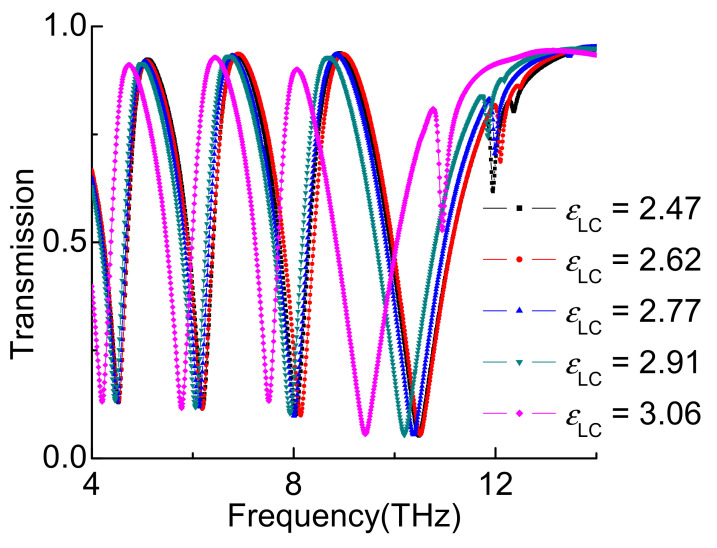
Transmission spectrum with different *ε*_LC_.

**Figure 5 nanomaterials-11-02793-f005:**
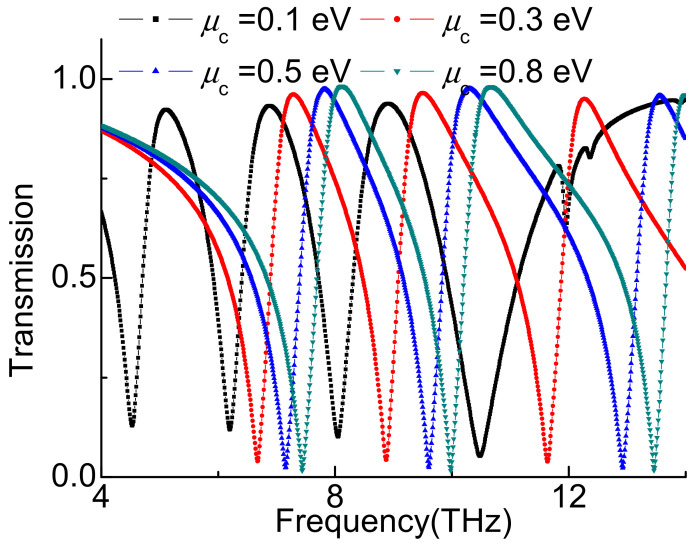
Transmission spectrum with different voltage of the graphene.

**Figure 6 nanomaterials-11-02793-f006:**
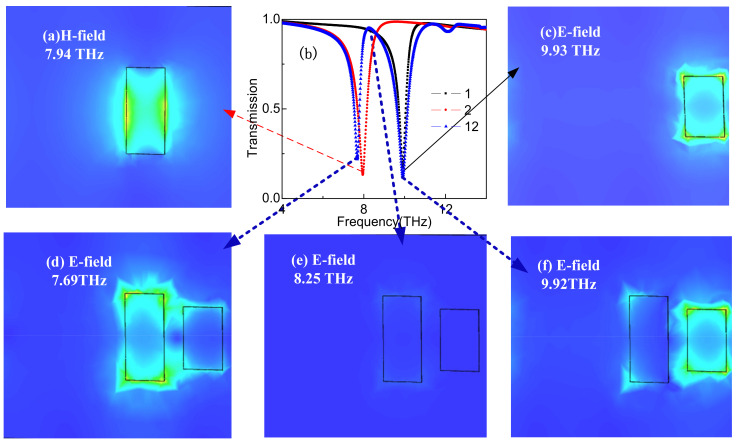
(**a**) magnetic field distribution, (**b**) The transmission spectra, and (**c**–**f**) electric field distribution of transmission spectra on normal incidence.

**Figure 7 nanomaterials-11-02793-f007:**
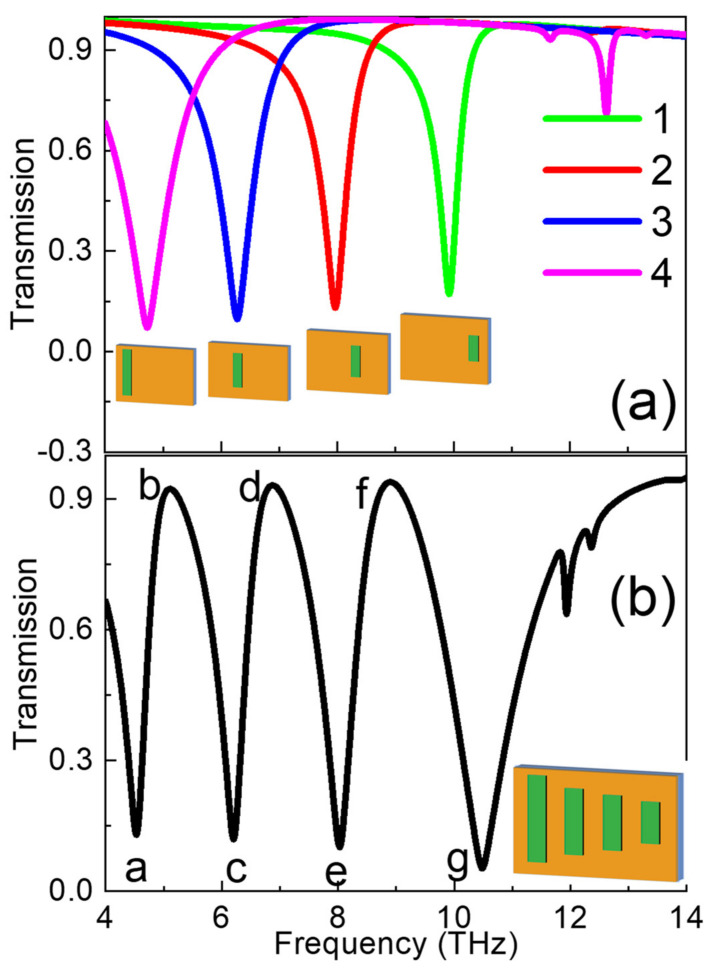
(**a**) Compared transmission spectra with different length graphene strips and (**b**) the whole structure.

**Figure 8 nanomaterials-11-02793-f008:**
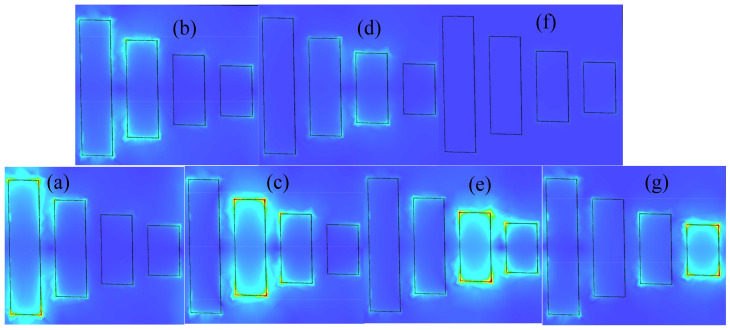
(**a**,**c**,**e**,**g**), (**b**,**d**,**f**) The electric field distribution of transmission spectra valleys of (**a**,**c**,**e**,**g**), transmission spectra peaks (**b**,**d**,**f**) in Figure 7b on normal incidence, respectivity.

**Figure 9 nanomaterials-11-02793-f009:**
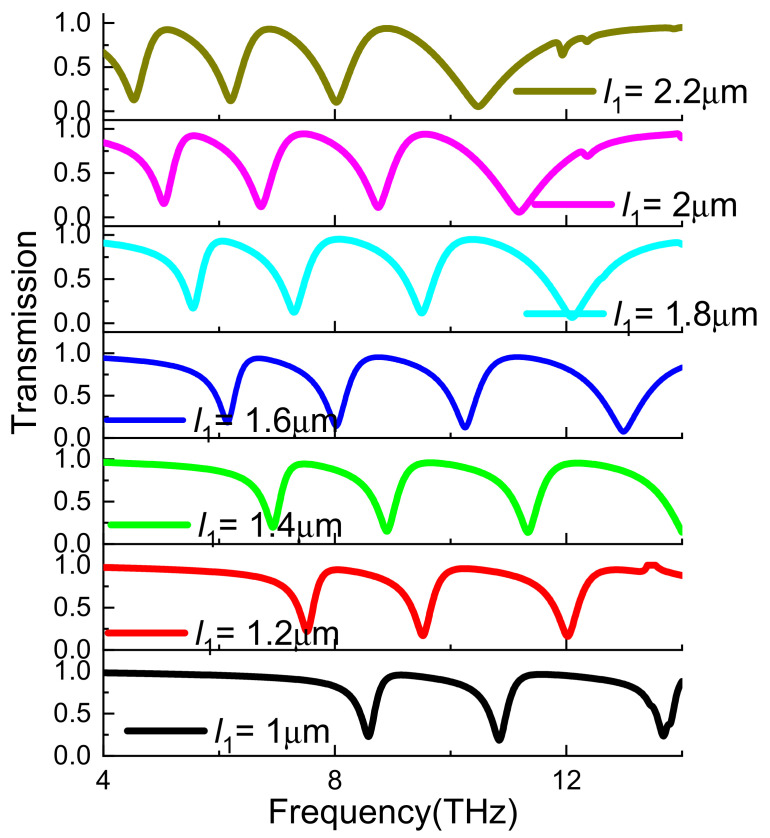
Simulated transmission spectra of the proposed A-EIT structures with respect to the length of the graphene strip.

**Figure 10 nanomaterials-11-02793-f010:**
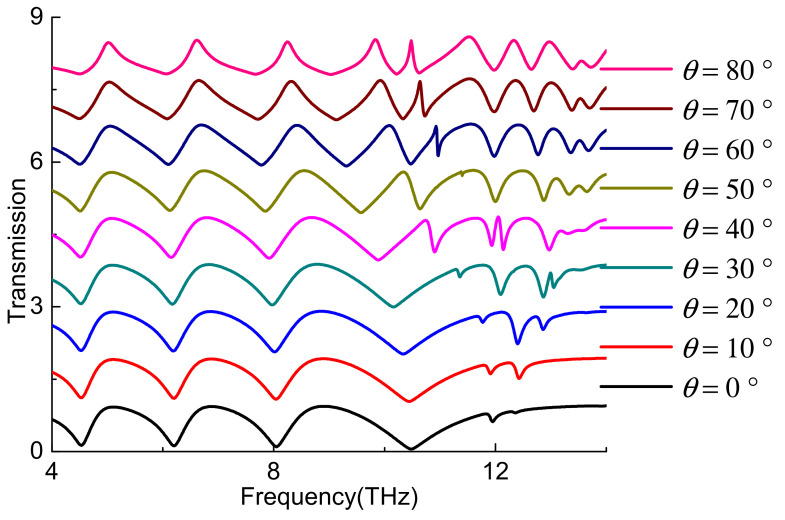
The transmission spectra under oblique incident angle *θ*.

**Figure 11 nanomaterials-11-02793-f011:**
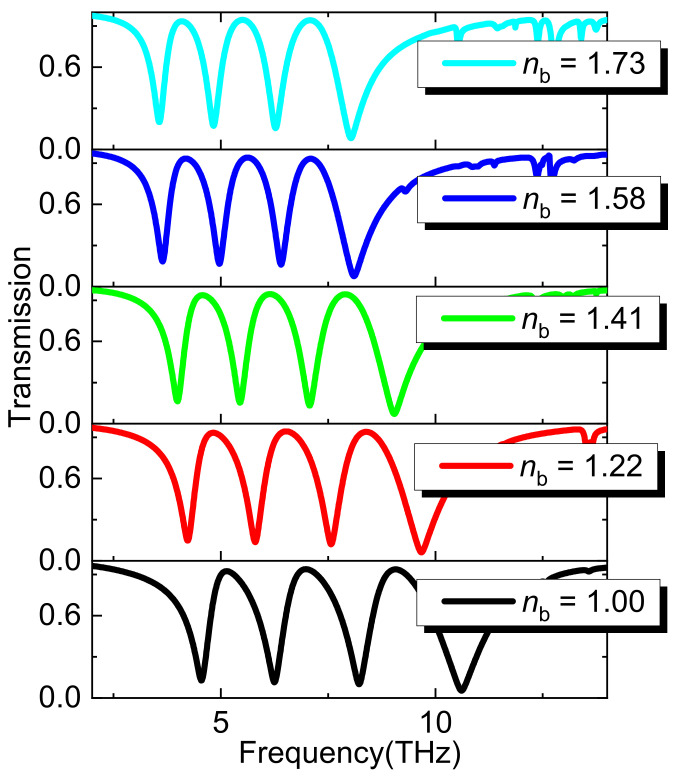
Transmission spectrum at different refractive index *n* of the surrounding environment.

**Figure 12 nanomaterials-11-02793-f012:**
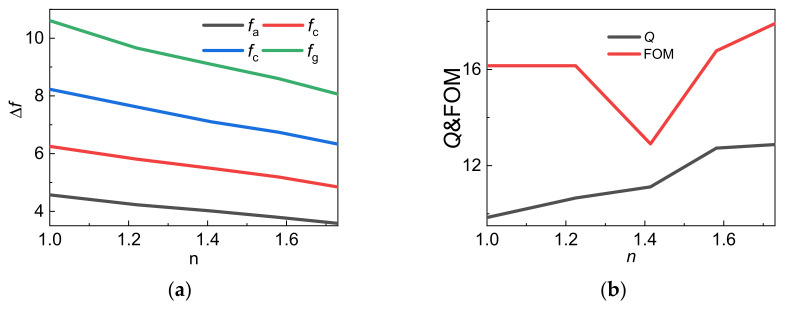
(**a**) Frequency shifting of *f*_a_, *f*_c_, *f*_e_, *f*_g_ on *n* from 1.0 to 1.73, (**b**) FOM and Q varied with *n* from 1.0 to 1.73.

**Table 1 nanomaterials-11-02793-t001:** Comparison of performance reported in various multiband A-EIT metamaterial.

Ref.	[28]	[29]	[30]	[31]	[32]
Band range	THz	THz	GHz	THz	GHz
material	Au	Dirac semimetal	Copper	Graphene	Copper
tunable	No	Single tunable	No	Single tunable	No
FOM	\	16, 50.9 and 9.6,	\	\	\
group delay	\	\	6.70, 0.76, 2.38 ns	\	1.1 and 0.7 ns
**Ref.**	**[33]**	**[34]**	**[35]**	**[9]**	**Our Work**
Band range	Optical	Optical	THz	THz	THz
material	ITO	Ag	Graphene	VO_2_	Graphene, LC
tunable	NO	NO	Single tunable	Single tunable	Dual-tunable
FOM	\	\	\	\	13, 16, 18
group delay	\	17, 19 ps	\	14 ps and 35 ps	\

## Data Availability

Not applicable.

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
