# Peer review of "Multi-Band Analogue Electromagnetically Induced Transparency in DoubleTuned Metamaterials"

_nanomaterials, 2021, doi:10.3390/nano11112793_

Round 1

Reviewer 1 Report

The authors of the presented manuscript present their theoretical project of an electrically-driven controllable metamaterial working in the THz range and based on a double tuning, related both to the compounding liquid crystal (LQ) and graphene layers, projected to achieve analogue EIT. The metamaterial design proceeds by an analytical representation of the graphene and LQ as a function of the applied electric voltages, and the resulting electromagnetic properties, mainly the transmissions spectra, are calculated by means of FDTD simulations after setting several control values.

The description here presented is clear and linear and evidently consistent with the investigation steps undertaken by the authors; it also possesses a discrete quality which helps to efficiently comprehend the entire work laying behind this project, despite some unavoidable defects on the English.

Unfortunately, there are also two main defects in it:

1) here we see a very deep lack of originality, since this manuscript addresses a widely probed topic in the literature of this area, with no substantial improvements and innovation, making a small and partial exception for the double tuning (which anyway does not add a real unique feature to this draft).
In few words, this work could be appreciable if used as a book chapter, because it presents a clear example of a tunable matematerial for controllable transmission, and can be inserted into a larger work adaptable to experimental developments and industrial applications. At this stage and with no further contributions, it represents a poor example of a metamaterial design with no true peculiarities and potentials.

2) this study is quite basic and does not investigate the effects of connecting the compounding layers on their same electromagnetic properties. For instance, the graphene is modeled as the contiguous layers do not affect it, while it is reasonable to expect a certain variation of its functionalities with respect to its stand-alone configuration.

I also address a minor request on this technical level. The resulting e.m. spectra are constrained to the weak coupling regime between graphene strips (see line 88, Chapter 2: Geometric Model); this coupling makes the overall structure easy to design and project, but it can be traced back to the setting of safety distances between strips, which in turn can lead to excessive space occupation of the overall structure and a consequent lack of efficiency. Therefore, the authors should complete their study by adding a special chapter, or at least a paragraph, to evaluate the EIT dependence on the spacing s parameter.

In sight of all of this, I consider this draft unfit to publication.

Minor issues are listed below (Unclear sentences deserving a recast, rough sentences with typos and/or unnecessary repetitions):

in the introduction, please add a technical definition of classical (i.e.  quantum) EIT, with its common and differential features with the analogue EIT,

Line 20, Abstract: and the A-EIT characteristics insensitive to the incident angle can be obtained (improve this sentence)

Line 23, Abstract: The multi-band A-EIT effect can widen the application scope of devices of the proposed EIT metamaterial on sensor and optical storage (improve this sentence)

Line 50, Introduction: EIT was first discovered in three-level atomic system, the conditions of the experiment are very strict. (this second part seems weakly connected with the former part)

Line 32, Introduction: in electromagnetic stealth, electromagnetic sensors and electromagnetic devices (avoid repetitions)

Line 40, Introduction: R. Kowerdziej et. al proposed a LC based on nematic liquid crystal (NLC) is …

Line 41, Introduction: on the electromagnetic characteristics (I suggest a different term, like properties)

Line 60, Introduction: The measurement result was shown a high quality factor transmitted peak can be observed (avoid inconsistencies by a recast on this statement)

Line 66, Introduction: The results show that its figure of merit (FOM) exceeding12.0 (either remove that or use exceeds)

Line 67, Introduction: The all-dielectric metasurface can dynamic manipulate EIT was investigated (recast this statement)

Line 95-96, Geometric Model: please, adopt a better representation for expression (4).

Line 100, Geometric Model: It is represented that the (recast)

Line 104, Geometric Model: The boundary condition of the structure is set as the unit cell boundary condition on x and y direction and open for the z-axis (Please, specify better x&y settings; are those set to PBC?)

Line 120, Geometric Model: to significant blue shifted (blue-shift).

Line 121, Geometric Model:  has the characteristics of double tunable (please, improve this statement)

Line 126, Geometric Model:  as bright modes can be excited by the incident wave at the same time. (this “same time” deserves a particular replacement or a explanation, because here we are inspecting data  drawn in linear regime and registered for distinct frequencies, so mixing time and frequency concepts requires particular cautions).

Line 127, Geometric Model:  The element of the metamaterial structure (I suggest to use “periodic unit” in place of “element”).

Line 134, Geometric Model:  It can be seen from Fig. 6 (d) that the electric field in the low frequency band is mainly distributed on the graphene strip 2 (I would recommend to use “field power” rather than “field”, in order to avoid further specifications of x-y field components, or to use different spectra for each component).

Line 136, Geometric Model:  indicating that transparency occurs at the frequency (please, complete this sentence).

Line 166, Geometric Model: at different resonance frequency (“frequencies” could be fitter that “frequency”. I recommend to specify which resonant frequency you are varying here, for instance by adding a reference to the parameters used in expr.(2,4,5)).

Author Response

Dear reviewer,  
Thank you for your letter and reviewer’ comments concerning our manuscript entitled “Multi-band Analogue Electromagnetically Induced Transparency in?Double Tuned Metamaterials” (ID: nanomaterials-1344854). Those comments are all valuable and very helpful for revising and improving our paper, as well as the important guiding significance to our researches. We have carefully studied the valuable comments and used them as guidance in our corrections. Revised portion are marked in red underline in the paper. The main corrections in the paper and the responds to the Reviewer’s comments are as flowing:

Reviewer comments:
The authors of the presented manuscript present their theoretical project of an electrically-driven controllable metamaterial working in the THz range and based on a double tuning, related both to the compounding liquid crystal (LQ) and graphene layers, projected to achieve analogue EIT. The metamaterial design proceeds by an analytical representation of the graphene and LQ as a function of the applied electric voltages, and the resulting electromagnetic properties, mainly the transmissions spectra, are calculated by means of FDTD simulations after setting several control values.
The description here presented is clear and linear and evidently consistent with the investigation steps undertaken by the authors; it also possesses a discrete quality which helps to efficiently comprehend the entire work laying behind this project, despite some unavoidable defects on the English.
Unfortunately, there are also two main defects in it:
1)here we see a very deep lack of originality, since this manuscript addresses a widely probed topic in the literature of this area, with no substantial improvements and innovation, making a small and partial exception for the double tuning (which anyway does not add a real unique feature to this draft).
In few words, this work could be appreciable if used as a book chapter, because it presents a clear example of a tunable matematerial for controllable transmission, and can be inserted into a larger work adaptable to experimental developments and industrial applications. At this stage and with no further contributions, it represents a poor example of a metamaterial design with no true peculiarities and potentials.
Answer: Thanks for the reviewer’s comment. In this paper, we design a simple structure to realize multi band EIT, and the results can be adjusted dynamically.?The purpose of this article is to provide an open idea for engineers and technicians.The structure has potential applications in refractive index sensors, slow light devices and large angle filters.?Moreover, taking silica as the substrate is not infeasible in the experimental scheme.Recently, we are focusing on the metamaterial structure of gallium nitride substrate, so this study can not be verified experimentally. Thank you again for your hard work.
2)this study is quite basic and does not investigate the effects of connecting the compounding layers on their same electromagnetic properties. For instance, the graphene is modeled as the contiguous layers do not affect it, while it is reasonable to expect a certain variation of its functionalities with respect to its stand-alone configuration.
Answer:Thank you very much for your comments.This paper studies the ideal electromagnetic characteristics of graphene without considering the interaction between adjacent layers.Here we assume a single-layer graphene. In the simulation results, the thickness of the single-layer graphene is usually set to 0(ACS Omega 2021, 6, 4480?4484),0.335nm(JOURNAL OF LIGHTWAVE TECHNOLOGY 2021, 39(5),1544-1549) and 1nm(IEEE JOURNAL OF SELECTED TOPICS IN QUANTUM ELECTRONICS, 2021, 27(1),4700406)
I also address a minor request on this technical level. The resulting e.m. spectra are constrained to the weak coupling regime between graphene strips (see line 88, Chapter 2: Geometric Model); this coupling makes the overall structure easy to design and project, but it can be traced back to the setting of safety distances between strips, which in turn can lead to excessive space occupation of the overall structure and a consequent lack of efficiency. Therefore, the authors should complete their study by adding a special chapter, or at least a paragraph, to evaluate the EIT dependence on the spacing?s?parameter.
Answer: Fig.1 shows that the transmission spectra varied with s at uc=0.5 eV. The results show that the change of S has an effect on the high frequency band and less on the low frequency band. It shows that s has little impact on A-EIT in this structure.

Fig.1 The transmission spectra under s.

In sight of all of this, I consider this draft unfit to publication.
Minor issues are listed below (Unclear sentences deserving a recast, rough sentences with typos and/or unnecessary repetitions):
in the introduction, please add a technical definition of classical (i.e.? quantum) EIT, with its common and differential features with the analogue EIT,
Line 20, Abstract: and the A-EIT characteristics insensitive to the incident angle can be obtained (improve this sentence)
Answer:Thank you very much for your comments.We improved this sentence “it is meaning that the A-EIT effect with insensitive to the incident angle can be obtained.”
Line 23, Abstract: The multi-band A-EIT effect can widen the application scope of devices of the proposed EIT metamaterial on sensor and optical storage (improve this sentence)
Answer:Thank you very much for your comments.The sentence have been improved “The multi-band A-EIT effect can widen the application on sensor and optical storage devices.”
Line 50, Introduction: EIT was first discovered in three-level atomic system, the conditions of the experiment are very strict. (this second part seems weakly connected with the former part)
Answer:Thank you very much for your comments.We removed this sentence. This paragraph and the previous paragraph describe EIT phenomena and EIT phenomena in metamaterials. We merged the two paragraphs into one.
Line 32, Introduction: in electromagnetic stealth, electromagnetic sensors and electromagnetic devices (avoid repetitions)
Answer:Thank you very much for your comments.We have been revised it. “It has potential applications in electromagnetic stealth devices, electromagnetic absorbers and electromagnetic sensors.”
Line 40, Introduction: R. Kowerdziej?et. al?proposed a LC based on nematic liquid crystal (NLC) is …
Answer:Thank you very much for your comments.We have corrected that”R. Kowerdziej et. al proposed a nematic liquid crystal (NLC) structure to determine the influence of alternating current (AC) bias voltage on the metamaterial sensor ”
Line 41, Introduction: on the electromagnetic characteristics (I suggest a different term, like?properties)
Answer:Thank you very much for your comments.We have revised it.
Line 60, Introduction: The measurement result was shown a high quality factor transmitted peak can be observed (avoid inconsistencies by a recast on this statement)
Answer:Thank you very much for your comments.We deleted the repetition part.
Line 66, Introduction: The results show that its figure of merit (FOM) exceeding12.0 (either remove?that?or use?exceeds)
Answer:Thank you very much for your comments.We deleted this sentence.
Line 67, Introduction: The all-dielectric metasurface can dynamic manipulate EIT was investigated (recast this statement)
Answer:Thank you very much for your comments.The sentence have revised “The all-dielectric metasurface structure was investigated which can be dynamic manipulated EIT effect in the optical telecommunication frequencies.”
Line 95-96, Geometric Model: please, adopt a better representation for expression (4).
Answer:Thank you very much for your comments.We have adjusted it.
Line 100, Geometric Model: It is represented that the (recast)
Answer:Thank you very much for your comments.It is revised that “It is shown that the real part of εG increases with the decrease of μc. “
Line 104, Geometric Model: The boundary condition of the structure is set as the unit cell boundary condition on x and y direction and open for the z-axis (Please, specify better x&y settings; are those set to PBC?)
Answer:Thank you very much for your comments.The boundary condition of x and y directions are unit cell and the z-axis is open.
Line 120, Geometric Model: to significant blue shifted (blue-shift).
Answer:Thank you very much for your comments. We have revised it.
Line 121, Geometric Model: ?has the characteristics of double tunable (please, improve this statement)
Answer:Thank you very much for your comments. We have revised that “It can be concluded that the structure has the double tunable property.”
Line 126, Geometric Model: ?as bright modes can be excited by the incident wave at the same time. (this “same time” deserves a particular replacement or a explanation, because here we are inspecting data ?drawn in linear regime and registered for distinct frequencies, so mixing time and frequency concepts requires particular cautions).
Answer:Thank you very much for your comments.We have modified the sentence. 
Line 127, Geometric Model: ?The element of the metamaterial structure (I suggest to use “periodic unit” in place of “element”).
Answer:Thank you for your comments. We have altered it.
Line 134, Geometric Model: ?It can be seen from Fig. 6 (d) that the electric field in the low frequency band is mainly distributed on the graphene strip 2 (I would recommend to use “field power” rather than “field”, in order to avoid further specifications of x-y field components, or to use different spectra for each component).
Answer:Thank you for your comments.We have revised this sentence.
Line 136, Geometric Model: ?indicating that transparency occurs at the frequency (please, complete this sentence).
Answer:Thank you very much for your comments.We have completed this sentence “indicating that transparency occurs at the frequency of 8.25 THz.”
Line 166, Geometric Model: at different resonance frequency (“frequencies” could be fitter that “frequency”. I recommend to specify which resonant frequency you are varying here, for instance by adding a reference to the parameters used in expr.(2,4,5)).
Answer:Thank you for your comments. We have revised the “frequencies” to “frequency”.
We tried our best to improve the manuscript and made some changes in the paper.??These changes will not influence the content and framework of the paper. We did not list the changes but marked in red underline in revised paper.
We appreciate your warm work, and hope that the correction will meet with approval.
Once again, thank you very much for your comments and suggestions.
I look forward to hearing from you soon.

Yours sincerely,
Renxia Ning

Reviewer 2 Report

In this manuscript, a simulation of multi-band analogue electromagnetically induced transparency effect was carried out on a metamaterial structure consisting of LC (liquid crystal)-dielectric (SiO2)-graphene strips layer. It seems innovative or significant for some new phenomena, however, to definitely description is highly necessary. Theoretical relations of the THz transmission with the characteristic sizes of the metamaterial structure, incident angle or the applied electrical voltage, should be more explicit. Further discussions on the mechanism or reasons behind the phenomena are always needed. It is hard to believe the validity and the veracity of this work at present submission.

Author Response

Dear reviewer,  

Thank you for your letter and reviewer’ comments concerning our manuscript entitled “Multi-band Analogue Electromagnetically Induced Transparency in?Double Tuned Metamaterials” (ID: nanomaterials-1344854). Those comments are all valuable and very helpful for revising and improving our paper, as well as the important guiding significance to our researches. We have carefully studied the valuable comments and used them as guidance in our corrections. Revised portion are marked in red underline in the paper. The main corrections in the paper and the responds to the Reviewer’s comments are as flowing:

In this manuscript, a simulation of multi-band analogue electromagnetically induced transparency effect was carried out on a metamaterial structure consisting of LC (liquid crystal)-dielectric (SiO2)-graphene strips layer. It seems innovative or significant for some new phenomena, however, to definitely description is highly necessary. Theoretical relations of the THz transmission with the characteristic sizes of the metamaterial structure, incident angle or the applied electrical voltage, should be more explicit. Further discussions on the mechanism or reasons behind the phenomena are always needed. It is hard to believe the validity and the veracity of this work at present submission.

Answer:Thank you for your comment. In this paper, we design a simple structure to realize multi band EIT, and the results can be adjusted dynamically. A-EIT effect is produced by the coupling of magnetic resonance and electric resonance is shown in FIg.6 (in the paper). The purpose of this article is to provide an open idea for engineers and technicians.The structure has potential applications in refractive index sensors, slow light devices and large angle filters. We added explanation to explain the influence of angle change on transmission spectrum.Once again thank the reviewer for your hard work.

We tried our best to improve the manuscript and made some changes in the paper.??These changes will not influence the content and framework of the paper. We did not list the changes but marked in red underline in revised paper.

We appreciate your warm work, and hope that the correction will meet with approval.

Once again, thank you very much for your comments and suggestions.

I look forward to hearing from you soon.

Yours sincerely,

Renxia Ning

Round 2

Reviewer 2 Report

Authors have revised the manuscript with their best effort,  it becomes acceptable now.